# Healthy Dads, Healthy Kids UK, a weight management programme for fathers: feasibility RCT

Tania Griffin,[1] Yongzhong Sun,[2] Manbinder Sidhu,[3] Peymane Adab,[4] Adrienne Burgess,[5] Clare Collins,[6] Amanda Daley,[7] Andrew Entwistle,[8] Emma Frew,[9] Pollyanna Hardy,[2] Kiya Hurley,[10] Laura Jones ,[4] Eleanor McGee,[11] Miranda Pallan,[4] Myles Young,[12] Philip Morgan,[12] Kate Jolly[4]

For numbered affiliations see end of article.

**Correspondence to**
Professor Kate Jolly;
c.b.jolly@bham.ac.uk

## ABSTRACT

**Objective** To assess (1) the feasibility of delivering a culturally adapted weight management programme, Healthy Dads, Healthy Kids United Kingdom (HDHK-UK), for fathers with overweight or obesity and their primary school-aged children, and (2) the feasibility of conducting a definitive randomised controlled trial (RCT).

**Design** A two-arm, randomised feasibility trial with a mixed-methods process evaluation.

**Setting** Socioeconomically disadvantaged, ethnically diverse localities in West Midlands, UK.

**Participants** Fathers with overweight or obesity and their children aged 4–11 years.

**Intervention** Participants were randomised in a 1:2 ratio to control (family voucher for a leisure centre) or intervention comprising 9 weekly healthy lifestyle group sessions.

**Outcomes** Feasibility of the intervention and RCT was assessed according to prespecified progression criteria: study recruitment, consent and follow-up, ability to deliver intervention, intervention fidelity, adherence and acceptability, weight loss, using questionnaires and measurements at baseline, 3 and 6 months, and through qualitative interviews.

**Results** The study recruited 43 men, 48% of the target sample size; the mean body mass index was 30.2 kg/m$^2$ (SD 5.1); 61% were from a minority ethnic group; and 54% were from communities in the most disadvantaged quintile for socioeconomic deprivation. Recruitment was challenging. Retention at follow-up of 3 and 6 months was 63%. Identifying delivery sites and appropriately skilled and trained programme facilitators proved difficult. Four programmes were delivered in leisure centres and community venues. Of the 29 intervention participants, 20 (69%) attended the intervention at least once, of whom 75% attended ≥5 sessions. Sessions were delivered with high fidelity. Participants rated sessions as 'good/very good' and reported lifestyle behavioural change. Weight loss at 6 months in the intervention group (n=17) was 2.9 kg (95% CI −5.1 to −0.6).

**Conclusions** The intervention was well received, but there were significant challenges in recruitment, programme delivery and follow-up. The HDHK-UK study was not considered feasible for progression to a full RCT based on prespecified stop–go criteria.

**Trial registration number** ISRCTN16724454.

## Strengths and limitations of this study

► This unique study provides new evidence into the feasibility of delivery of a weight management programme targeting fathers from socioeconomically deprived, ethnically diverse community settings, and its acceptability in this target population, a key strength given the relative paucity of research in this area.

► Recruitment methods were wide ranging, raising awareness of the study in a number of community settings; however, despite extensive efforts, recruitment to the study was a key challenge.

► The intention was to recruit male facilitators to deliver the intervention sessions. However, it was difficult to identify suitable facilitators with the necessary skill set; therefore, a range of male and female facilitators were recruited.

► Identifying a convenient time to run the intervention course to suit fathers, venue and facilitator availability proved challenging.

## INTRODUCTION

Overweight and obesity remain a public health challenge and priority.[1] At an individual level, excess weight results in fewer disease-free years and lowered life expectancy,[2 3] and at a population level, obesity prevalence is associated with major economic burden.[4]

In England, overweight and obesity levels are higher in men compared with women with socioeconomic inequalities evident.[5] Men from the lowest income quintile have a greater mean waist circumference than men from the highest income quintile.[6] South Asian men have higher body fat percentage than white Europeans at the same body mass index (BMI).[7]

The transition to fatherhood has been identified as a critical period during which men are more likely to adopt obesogenic behaviours and are more susceptible to

weight gain.[8 9] While there have been several successful men-only weight management programmes,[10–14] attendance to such activities remains higher among women.[13] Fathers are also under-represented in interventions targeting family health behaviours.[15] This emphasises the need to identify programmes which fathers are keen to engage with.

Healthy Dads, Healthy Kids (HDHK) is a weight management programme for fathers of primary school-aged children that also addresses the healthy eating and physical activity behaviours of their children. The programme, developed and delivered in Australia, was shown to be successful at achieving weight loss in fathers.[16 17] In the current study, the HDHK programme was adapted with the intention of making it culturally acceptable to a multiethnic UK population.[18] The aim of this study was to assess the feasibility of delivering the adapted HDHK programme and the feasibility to conduct a future randomised controlled trial (RCT) in a socioeconomically deprived ethnically diverse UK setting.

## METHODS
### Study design and participants
This study was a two-arm, randomised feasibility trial with a mixed-methods process evaluation. It was conducted in two urban local authority areas of the West Midlands, UK (site 1 and site 2) selected for their population profile and interest in supporting the programme. In 2017, both areas were ranked in the most deprived 20% of areas in the UK, with high ethnic diversity.[19]

The study aimed to recruit men who were father/stepfather/father figures (herein referred to as 'fathers') of primary school children (aged 4–11 years), aged 18–65 years with a BMI of ≥25 kg/m$^2$ (≥23 kg/m$^2$ for minority ethnic groups)[7] and/or a waist circumference of ≥94 cm (37 inches) who were willing to lose weight. Fathers did not have to be coresident with their children. Men were not eligible if they had a history of angina or other cardiovascular disease; had orthopaedic or joint problems that would be a barrier to vigorous physical activity; had weight loss of 3 kg/7 lb in the previous 3 months; had current diabetes and were not confident in managing their condition during exercise; were unable to speak and/or understand English; and were involved in ongoing custody or access disputes and/or any contexts with a risk of domestic violence.

### Sample size
As a feasibility study, no formal sample size calculation was performed. We aimed to recruit 90 men and their children to estimate the recruitment, follow-up and questionnaire completion rates to within ±10% with 95% CIs, based on a worst case estimate of 50%.

### Recruitment and randomisation
Fathers were recruited (September 2017–January 2018) by the research team who had extensive experience of participant recruitment in a community setting. A range of methods were used over the recruitment period, including flyer distribution and promotion stands at leisure, community and shopping centres, places of worship and large workplace organisations. Recruitment via schools conducted through presentations at school assemblies and teacher meetings, stands at parent evenings, flyer distribution and talking to parents at school pick-up time. The study was promoted on social media (Twitter and Facebook).

Written informed consent was obtained from fathers and assent from children aged 8 years and over. Once baseline data were collected, participants were randomised (1:2 allocation ratio) to the control group (voucher for a single family visit to a leisure centre) or intervention group (adapted HDHK-UK programme). Randomisation was stratified by the father's ethnicity (white British or Irish/other ethnic group) and conducted using an automated online form developed by the University of Birmingham Clinical Trials Unit.

### Intervention
The HDHK-UK intervention comprised weekly 90 min sessions over nine consecutive weeks; four courses were delivered. Fathers and children attended all sessions, which followed the same structure: 15 min discussion and review of the weekly activities followed by 30 min, where children and fathers took part in an education session separately. The groups were facilitated by local, experienced and trained staff to ensure the sessions were interactive and discussion was encouraged. Facilitators were selected based on their experience of delivering group programmes and delivering health advice; they included health trainers (who provide community support for health-related behaviour changes) and sports coaches. They completed HDHK delivery training with either the Fatherhood Institute or the research team. The training included practising delivery of parts of the intervention. Fathers' sessions covered a range of lifestyle behaviours around the importance of physical activity, nutrition and parenting. Children were taught about healthy eating, physical activity and how to be a supportive family member by encouraging and modelling healthy lifestyle behaviours at home. The final 45 min of the session were spent doing physical activity within family groups. These practical sessions had three elements: 'rough and tumble' play; teaching children fundamental movement skills (catching, throwing and kicking); and aerobic fitness. The intervention is summarised in the template for intervention description and replication (TIDieR) checklist (online supplementary table 1). The HDHK intervention draws on concepts from family systems theory[20 21] and social cognitive theory[22]; the theoretical constructs are reported for the original programme.[17] Adaptations to the Australian resources were made by the research team (KJ and MSS) and one of the study partners (the Fatherhood Institute) in conjunction with the wider research team. The adaptations were informed by qualitative research

with fathers and mothers from similar socioeconomic backgrounds and residing in the same geographical region.[23] Adaptations focused on reducing the number of PowerPoint slides, simplifying and anglicising wording and updating the guidance and statistics to align to UK public health recommendations. References to foods, activities and images were updated to reflect a multicultural UK population. Fathers and their children attended every session and mothers were invited to one session when family food was discussed.

## Data collection and outcomes

Data were collected from participants by trained researchers at baseline and at 3 and 6 months later in their home (or convenient location). Sociodemographic characteristics were collected by questionnaire. Based on postal code of residence, the Index of Multiple Deprivation (IMD) was used to determine socioeconomic status.[24]

### Primary outcomes: feasibility measures

The outcomes relating to feasibility of delivery of HDHK-UK were the ability to recruit and retain facilitators, the ability to deliver sessions at a time and location convenient for participants, fidelity of delivery and acceptability to participants.

The outcomes relating to the feasibility of conducting a future definitive RCT were recruitment rate, willingness to be randomised, follow-up rates, level of completion of follow-up questionnaires, and father's weight change in the intervention group.

### Process evaluation

Process measures were collected by the research team to determine the feasibility of intervention delivery and study processes, fidelity of intervention delivery and participant acceptability. Sixteen session observations (minimum of two per intervention course) were completed to assess content delivery and participant engagement. Participants and facilitators were also asked to complete feedback forms after each session and evaluating the session content and delivery. Qualitative interviews were conducted with 12 participants postintervention. One participant was interviewed at both 3 and 6 months. The average interview duration was 16 min, and interviews were conducted either face-to-face (n=4) or by telephone (n=9). All facilitators (n=7) were interviewed; five interviews were conducted individually, one with two facilitators together.

### Secondary outcome measures

Secondary outcome measures for fathers were outcomes that would be collected in the main trial (if progressed to). These comprised percentage losing ≥5% body mass, change in waist circumference and percent body fat, self-reported physical activity (using the IPAQ-short,[25] objectively measured physical activity (by a wrist-worn GeneActiv (Activinsights, Cambs, UK), triaxial accelerometer worn for 7 days on the father's non-dominant wrist), self-reported dietary intake (using food frequency items

(Food Frequency Questionnaire (FFQ)), father–child relationship outcomes (using the Parent–Child Relationship Questionnaire),[26] parenting for physical activity (using the Parenting Strategies for Eating and Activity Scale[26] and the Physical Activity Modelling Subscale of the Activity Support Scale for Multiple Groups.[27]

Secondary outcome measures collected from the children comprised BMI z-score change (calculated using the LMS method[28] and UK reference data[29]); per cent body fat, overweight or obese, objectively measured physical activity (GeneActiv, worn by the eldest child only); parent-reported dietary intake for the eldest child (using the Family Nutrition and Physical Activity Questionnaire[30]); and behaviour and emotional well-being (using the Strength and Difficulties Questionnaire).[31]

To assess the feasibility of collecting outcome data for a future health economic analysis, we collected the EQ-5D-5L[32] and ICEpop CAPability Measure for Adults[33] from fathers and the Child Health Utility-9D[33–35] from the children.

## Progression criteria

Progression criteria, agreed on by the funding panel and the Study Steering Committee, were predefined to help evaluate whether the feasibility trial should be recommended to progress to a fully powered RCT. These are detailed in the Results section.

## Data analysis

For the quantitative analyses, the aim was to assess the progression criteria and the feasibility of delivery of a main trial. All analyses were by intention-to-treat and were undertaken in STATA V.12. Feasibility outcomes are presented overall and by group with counts, percentages and 95% CIs. Weight changes in fathers at follow-up of 3 and 6 months are summarised using means, SD and within-group 95% CIs. Baseline characteristics are summarised by group and overall, and the proposed secondary outcomes of a definitive trial are summarised by group only. Categorical data are presented using counts and percentages, and continuous data are presented using means and SDs or medians and IQRs, where appropriate. Since the study was not powered to detect treatment effects on clinical outcomes, p values and 95% CIs were not reported.

Audio recordings of qualitative interviews were transcribed verbatim, anonymised and analysed using the framework approach.[36]

## Public involvement

A public and patient participation group of eight fathers and two mothers from one of the research sites was involved throughout all stages of the study by contributing to decisions about outcome measures, commenting on intervention materials (in particular, the need for their simplification) and participant facing documents (downplaying the focus on weight management and focussing on the opportunity for father–child interaction

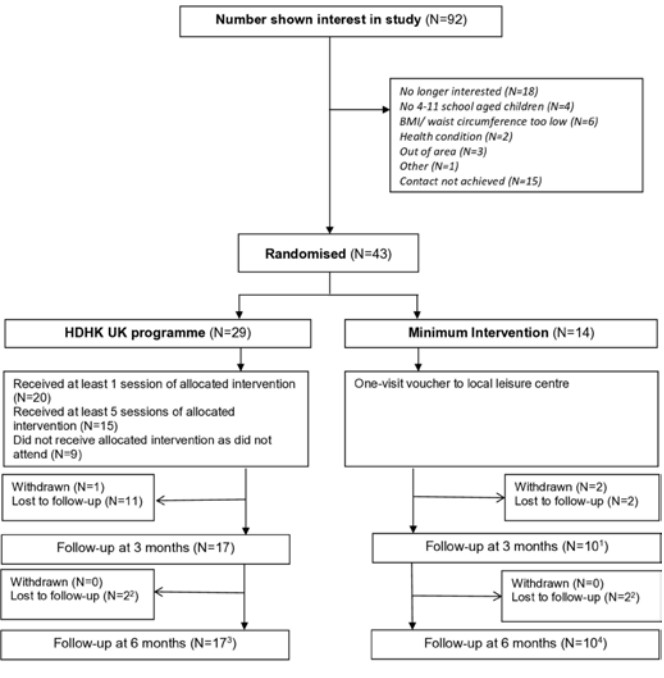

¹ One participant completed questionnaire only;
² Lost to follow up out of those participants who completed follow up at month 3;
³ 15 participants continued from the previous 17 who completed the follow up at month 3; 2 participants returned from lost to follow up at month 3;
⁴ 8 participants continued from the 10 who completed follow up at month 3, 2 participants returned from lost to follow up at month 3.

**Figure 1** Participant flow through HDHK-UK for fathers. BMI, body mass index; HDHK-UK, Healthy Dads, Healthy Kids United Kingdom.

in recruitment materials), and advising on recruitment strategies and dissemination of the findings.

## RESULTS
### Feasibility of conducting a definitive RCT

Despite multiple recruitment methods, only 92 men expressed interest in the study, of whom 43 were recruited and randomised (intervention group, n=29) (figure 1); 62 children were recruited. No participants declined randomisation after an assessment visit, nor was randomisation given as a reason for declining after expressing an interest in the study.

Four HDHK-UK courses were delivered, one at site 1 (in a youth centre) and three at site 2 (two in a leisure centre and one in a community centre).

### Baseline characteristics of participants

The fathers' mean BMI at baseline was $30.2\,\text{kg/m}^2$ (SD 5.1), and their mean age was 40.0 years (range 23.8–56.0). The mean age of their participating children (n=62) was 7.7 years (range 4.0–11.7), of whom 20 (32.8%) were overweight or obese. Overall, 60.5% of the participants were from minority ethnic groups, and 74.4% lived in the two most deprived quintiles of the IMD. Details of the baseline characteristics are presented in table 1.

### Follow-up rates

The follow-up rate at 3 and 6 months postintervention was 62.8% (n=27): 58.6% (n=17) in the intervention group

and 71.4% (n=10) in the control group (figure 1). Participants lost to follow-up at 6 months were more likely to be white British (online supplementary table 2).

### Level of completion of follow-up questionnaires

Researchers experienced challenges in arranging appointments for data collection. Despite repeated reminders, researchers often arrived at a family's house to find they were no longer available. Work and after-school activities made finding times for recruitment visits difficult and impacted recruitment numbers. For fathers and their children who completed data collection appointments, the processes worked well. In the main, the completeness of the data was acceptable with the exception of high levels of missing data for the IPAQ questionnaire and fathers' waist circumference, which was frequently refused (23% missing at baseline).

### Fathers' weight change in the intervention group

Table 2 summarises fathers' weight change from baseline. The intervention group had a 2.9 kg (95% CI 0.6 to 5.1) reduction in weight at 6 months' follow-up.

### Secondary outcomes

Secondary outcomes for fathers are summarised in table 2, and those for children are summarised in table 3. The intervention group of fathers had favourable reduction in waist circumference; 31% (n=9) achieved a 5% reduction in body mass at 6 months. There were no serious adverse events requiring hospital admission or adverse events requiring medical attention during the intervention.

### Feasibility and acceptability of programme delivery
### Ability to recruit and retain facilitators

Site 1 provided health trainers employed by the local authority for the fathers' education component, and an independent fitness trainer delivered the physical activity elements. At site 2, two courses were delivered in a leisure centre by their staff, and one course was delivered in a community centre by sports coaches employed by a coaching organisation.

We experienced challenges in delivering training due to loss of potential facilitators as a result of rapid turnover of staff from various delivery organisations. Three forms of training were delivered: a 2-day HDHK training workshop delivered by the Fatherhood Institute, which had previously been conducted by the Australian team; a half day of top-up training by the Fatherhood Institute; and more flexible training for facilitators recruited after these sessions, delivered by a member of the research team (TLG) who had previous experience in coaching training.

We intended that two courses would be delivered at each site. However, due to organisational restructuring, the health trainers were only able to deliver one course.

### Challenges to recruiting participants

There were significant challenges with recruitment. We obtained permission from 40 organisations to recruit

**Table 1** Baseline characteristics for fathers and children by treatment arm

| | HDHK-UK programme | Minimum intervention | Overall |
|---|---|---|---|
| | | | **Overall** |
| Fathers | n=29 | n=14 | N=43 |
| Age (years) | | | |
| Mean (SD), N | 39.4 (6.3), 29 | 41.1 (6.6), 14 | 40.0 (6.4), 43 |
| Ethnicity | | | |
| White British | 12 (41.4) | 5 (35.7) | 17 (39.5) |
| Non-white British | 17 (58.6) | 9 (64.3) | 26 (60.5) |
| Highest level of qualification | | | |
| GCSE, CSE, O level or equivalent | 7 (24.1) | 6 (42.9) | 13 (30.2) |
| A level/AS level or equivalent | 3 (10.3) | 2 (14.3) | 5 (11.6) |
| Degree level or higher | 15 (51.7) | 6 (42.9) | 21 (48.8) |
| Other | 2 (6.9) | 0 (0.0) | 2 (4.7) |
| Missing | 2 (6.9) | 0 (0.0) | 2 (4.7) |
| Legal marital or civil partnership status | | | |
| Married or in a registered civil partnership | 25 (86.2) | 13 (92.9) | 38 (88.4) |
| Divorced or formerly in a civil partnership, which is now legally dissolved | 1 (3.5) | 0 (0.0) | 1 (2.3) |
| Never married and never registered in a civil partnership | 1 (3.5) | 1 (7.1) | 2 (4.7) |
| Missing | 2 (6.9) | 0 (0.0) | 2 (4.7) |
| Main spoken language | | | |
| English | 23 (79.3) | 14 (100.0) | 37 (86.1) |
| Urdu | 1 (3.5) | 0 (0.0) | 1 (2.3) |
| Punjabi | 1 (3.5) | 0 (0.0) | 1 (2.3) |
| Spanish | 1 (3.5) | 0 (0.0) | 1 (2.3) |
| Turkish | 1 (3.5) | 0 (0.0) | 1 (2.3) |
| Missing | 2 (6.9) | 0 (0.0) | 2 (4.7) |
| Index of multiple deprivation quintile, n (%) | | | |
| 1 (least deprived) | 1 (3.5) | 0 (0.0) | 1 (2.3) |
| 2 | 1 (3.5) | 1 (7.1) | 2 (4.7) |
| 3 | 5 (17.2) | 1 (7.1) | 6 (14.0) |
| 4 | 5 (17.2) | 4 (28.6) | 9 (20.9) |
| 5 (most deprived) | 17 (58.6) | 6 (42.9) | 23 (53.5) |
| Missing | 0 (0.0) | 2 (14.3) | 2 (4.7) |
| BMI (kg/m²) | | | |
| Mean (SD), n | 30.1 (4.8), 29 | 30.2 (5.8), 14 | 30.2 (5.1), 43 |
| Weight at baseline (kg) | | | |
| Mean (SD), n | 90.1 (13.2), 29 | 92.2 (19.3), 14 | |
| Percentage body fat (%) | | | |
| Mean (SD), n | 28.3 (7.6), 28 | 29.8 (9.0), 14 | 28.8 (8.0), 42 |
| Waist circumference (cm) | | | |
| Mean (SD), n | 101.8 (8.5), 19 | 103.1 (15.0), 14 | 102.3 (11.5), 33 |
| International physical activity questionnaire[25] (IPAQ-short) | | | |
| Low activity | 8 (27.6) | 6 (42.9) | 14 (32.6) |
| Moderate activity | 10 (34.5) | 4 (28.6) | 14 (32.6) |
| Vigorous activity | 10 (34.5) | 3 (21.4) | 13 (30.2) |
| Missing | 1 (3.5) | 1 (7.1) | 2 (4.7) |
| Total activity (min) from GENEactiv measurement | | | |

| Table 1  Continued | | | |
| --- | --- | --- | --- |
| | **HDHK-UK programme** | **Minimum intervention** | **Overall** |
| Median (IQR) | 207.3 (175.8–270.1) | 213.6 (157.4–248.9) | 210.4 (167.7–264.6) |
| Missing | 4 | 1 | 5 |
| Moderate/vigorous activity (min) from GENEactiv measurement | | | |
| Median (IQR) | 110.8 (88.4–141.3) | 87.0 (75.9–148.4) | 109.7 (83.8–148.4) |
| Missing | 4 | 1 | 5 |
| EQ-5D-5L Index score[32] (adult quality of life) | | | |
| Mean (SD) | 0.91 (0.16) | 0.95 (0.07) | 0.93 (0.13) |
| Missing | 2 | 0 | 2 |
| ICECAP-A total capability score[44] (adult well-being and quality of life) | | | |
| Mean (SD) | 0.86 (0.15) | 0.95 (0.06) | 0.89 (0.13) |
| Missing | 1 | 0 | 1 |
| Children | n=42 | n=19 | N=61 |
| Age (years) | | | |
| Mean (SD) | 7.7 (2.0) | 7.8 (2.2) | 7.7 (2.1) |
| Minimum–maximum | 4.2–11.5 | 4.0–11.7 | 4.0–11.7 |
| Sex | | | |
| Female | 16 (38.1) | 6 (31.6) | 22 (36.1) |
| Male | 26 (61.9) | 13 (68.4) | 39 (63.9) |
| BMI (kg/m²) | | | |
| Underweight/healthy | 20 (47.6) | 13 (68.4) | 33 (54.1) |
| Overweight/obese | 16 (38.1) | 4 (21.1) | 20 (32.8) |
| Missing | 6 (14.3) | 2 (10.5) | 8 (13.1) |
| Percentage body fat (%) | | | |
| Mean (SD) | 24.0 (5.8) | 22.9 (4.9) | 23.6 (5.5) |
| Minimum–maximum | 10.3–36.7 | 16.9–37.3 | 10.3–37.3 |
| Missing | 6 | 3 | 9 |

Note: All figures presented are n (%) unless otherwise specified.
A level, Advanced level; AS level, Advanced Subsidiary level; BMI, body mass index; CSE, Certificate of Secondary Education; EQ-5D-5L, EuroQol; GCSE, General Certificate of Secondary Education; HDHK-UK, Healthy Dads, Healthy Kids United Kingdom; ICECAP-A, ICEpop CAPability Measure for Adults; O level, Ordinary level.

on their premises (including 11 schools, 7 faith organisations and 6 sport/leisure centres), but it was a time-consuming process. Communication with schools was slow and recruitment via children and 'mothers' was not a successful strategy. Fathers were hard to engage during recruitment activities, and session timings were often not convenient for their family. Additionally, as a result of a change in sites after funding was awarded (due to a local site withdrawing funding for adult weight management services), the sites involved considerable researcher travel time.

### Deliver sessions at a time and location convenient for participants

Aligning venue, facilitator and participant availability was a major challenge and resulted in considerable delays in commencing programme delivery. Evening sessions were difficult due to fathers' work commitments and children's meal and bed times. Although weekends were reported to be convenient by participants, the availability of facilitators and community venues was more limited.

> I wasn't finishing work until 5 o'clock and then I was having to fight my way through the traffic to get there. If it had been any later, it's harder for the kids then because they've got to get up for school the next week. … and weekends wouldn't be any good because the leisure centre would be packed. (ID B-008)

Session timings were also the main reason given by participants for non-attendance at intervention sessions.

> It started at half past five which is obviously rush hour time. Dads are leaving work … In the end, what happened was it went from a group of six dads and their children to the last session being just three dads ….

**Table 2** Weight change and secondary outcome measures for fathers by treatment arm

| | 3 months | | 6 months | |
|---|---|---|---|---|
| | HDHK-UK programme (n=29) | Minimum intervention (n=14) | HDHK-UK programme (n=29) | Minimum intervention (n=14) |
| Weight change from baseline (kg) | | | | |
| Mean (SD), n (95% CI) | −1.8 (2.5), 17 (−3.1 to −0.5) | −1.2 (3.3), 9 (−3.7 to 1.3) | −2.9 (4.1), 15 (−5.1 to −0.6) | −2.0 (3.6), 10 (−4.6 to 0.6) |
| Change from baseline in waist circumference (cm) | | | | |
| Mean (SD), n | −10.8 (18.0), 9 | 3.4 (9.3), 6 | −5.2 (5.0), 6 | −2.8 (6.4), 5 |
| Change from baseline in % body fat | | | | |
| Mean (SD), n | −1.5 (3.1), 16 | −0.3 (2.6), 9 | −2.2 (3.2), 14 | −2.3 (3.9), 9 |
| Physical activity measured by a GENEactive accelerometer | | | | |
| Median for total activity (min) (IQR), n | 208.8 (185.6–287.0), 17 | 168.3 (147.8–194.0), 9 | 239.6 (194.3–287.0), 11 | 146.2 (125.0–230.6), 9 |
| Median for moderate/vigorous activity (IQR), n | 113.3 (99.3–151.3), 17 | 84.2 (69.4–106.2), 9 | 125.6 (101.3–163.1), 11 | 68.2 (59.1–86.5), 9 |
| **EQ-5D-5L**[32] (adult health related quality of life) | | | | |
| Mean (SD), n | 0.87 (0.19), 17 | 0.94 (0.10), 9 | 0.92 (0.14), 14 | 0.94 (0.13), 9 |
| ICECAP-A[34] (adult capability and well-being) | | | | |
| Mean (SD), n | 0.92 (0.09), 17 | 0.95 (0.05), 9 | 0.89 (0.13), 14 | 0.92 (0.07), 9 |
| Lost ≥5% body mass | | | | |
| Yes | 2 (6.9%) | 1 (7.1%) | 9 (31.0%) | 1 (7.1%) |
| No | 15 (51.7%) | 8 (57.1%) | 6 (20.7%) | 9 (64.3%) |
| Missing | 12 (41.4%) | 5 (35.7%) | 14 (48.3%) | 4 (28.6%) |
| Parenting for physical activity | | | | |
| Activity Support Scale for Multiple Groups[27] | | | | |
| Mean (SD), n | 15.6 (3.5), 17 | 13.0 (2.3), 9 | 16.4 (2.2), 14 | 13.7 (1.5), 9 |
| Parenting Strategies for Eating and Activity Scale,[26] mean (SD), n | | | | |
| Limit setting | 8.4 (1.9), 17 | 8.8 (1.4), 9 | 7.5 (2.1), 13 | 9.1 (1.2), 9 |
| Control | 2.7 (1.6), 17 | 2.1 (1.3), 9 | 3.4 (1.3), 13 | 2.6 (1.7), 9 |
| Monitoring | 7.3 (1.5), 17 | 7.4 (1.1), 9 | 7.4 (1.5), 14 | 7.7 (0.7), 9 |
| Disciplining | 5.2 (2.3), 17 | 4.7 (2.4), 9 | 5.2 (2.7), 14 | 6.3 (2.2), 9 |
| Cophysical activity | 3.5 (0.9), 17 | 2.7 (0.7), 9 | 3.8 (1.3), 14 | 3.2 (0.8), 9 |
| Father–child relationship,[45] mean (SD), N | | | | |
| Disciplinary warmth* | 22.9 (4.0), 16 | 23.0 (1.6), 9 | 23.8 (3.6), 14 | 23.3 (3.0), 9 |
| Personal relationships† | 31.4 (5.0), 16 | 28.9 (2.6), 9 | 30.6 (4.8), 14 | 30.0 (2.9), 9 |

*Disciplinary warmth=praise + shared decision making +rationale
†Personal relationships (prosocial+intimacy+nurturance+companionship).
BMI, body mass index; EQ-5D-5L, EuroQol; ICECAP-A, ICEpop CAPability Measure for Adults.

I think that wasn't due to their motivation but their work time commitments. (ID B-068)

### Fidelity of delivery

The intervention was delivered with high fidelity. The style of delivery and the small group sizes meant group discussions were encouraged, allowing content to be tailored to the group's needs (facilitators were trained to manage group discussions as part of the HDHK delivery training); however, facilitators reported finding it challenging to deliver all the session contents in the allocated time. This was especially difficult if participants arrived to the sessions late and facilitators had to balance content delivery while also allowing for group discussion and interaction. Researchers sometimes observed some session content being skipped or a delay in the start of the subsequent practical session. Despite the challenges, the facilitators ensured key content was delivered, which was verified in participant interviews, where they spoke of the key messages the course had focused on.

**Table 3** Secondary outcome measures for eldest child by treatment arm

| | 3 months | | 6 months | |
|---|---|---|---|---|
| | HDHK-UK programme (n=29) | Minimum intervention (n=14) | HDHK-UK programme (n=29) | Minimum intervention (n=14) |
| Change from baseline in BMI z-score | | | | |
| Mean (SD), n | −0.131 (0.272), 14 | 0.016 (0.346), 8 | −0.016 (0.299), 12 | 0.039 (0.449), 10 |
| Change from baseline in % body fat | | | | |
| Mean (SD), n | −0.58 (1.17), 13 | 0.04 (2.84), 8 | −0.80 (1.48), 11 | −0.88 (3.07), 8 |
| Categorised as overweight or obese (%) | | | | |
| Underweight/healthy | 8 (27.6%) | 6 (42.9%) | 7 (24.1%) | 6 (42.9%) |
| Overweight/obese | 7 (24.1%) | 3 (21.4%) | 6 (20.7%) | 4 (28.6%) |
| Missing | 14 (48.3%) | 5 (35.7%) | 16 (55.2%) | 4 (28.6%) |
| Physical activity measured by a GENEactive accelerometer | | | | |
| Median for total activity (min) (IQR), n | 342.3 (262.7–427.7), 17 | 277.0 (272.5–314.3), 9 | 347.3 (321.6–384.0), 10 | 312.8 (245.4–456.8), 8 |
| Median for moderate/vigorous activity (IQR), n | 73.5 (34.7–99.3), 17 | 57.0 (26.8–73.0), 9 | 73.2 (49.0–105.7), 10 | 56.0 (32.4–110.6), 8 |
| Family nutrition and physical activity questionnaire[30] | | | | |
| Mean (SD), n | 62.3 (7.7), 16 | 61.1 (5.2), 9 | 61.2 (6.5), 9 | 62.1 (3.0), 8 |
| CHU-9D[33–35] (child utility measure) | | | | |
| Mean (SD), n | 0.89 (0.09), 15 | 0.93 (0.04), 9 | 0.92 (0.09), 11 | 0.92 (0.11), 8 |
| SDQ*[31] (total SDQ) | | | | |
| Mean (SD), n | – | – | 7.4 (3.8), 12 | 11.2 (6.1), 6 |

*SDQ: Each 1-point increase in the total difficulties score corresponds with an *increase in the risk* of developing a mental health disorder.
BMI, body mass index; CHU-9D, Child Health Utility-9D; SDQ, Strengths and Difficulties Questionnaire.

## Acceptability of adapted intervention to participants

Twenty participants (69%) attended an intervention course at least once; 15 attended (52%) at least five sessions. Fathers who attended the sessions and took part in a qualitative interview were positive about their experiences. Feedback sheets, observations and qualitative interviews with both participants and facilitators consistently showed high levels of acceptability towards the intervention programme.

It [HDHK-UK programme] was brilliant overall. I really enjoyed it. The kids enjoyed it. (ID B-068)

The group sessions were appreciated by participants:

I enjoyed the group-based elements … it allowed to people to bounce off, talk about - and also the blokes, a little bit of competitive edge, especially when the weight round was coming. So I think it benefited me and I preferred the group format rather than just individually between me and my children. (ID A-077)

A strong theme was the appreciation of time spent with their children: 'I'm working during the week and it's just nice to have that dad-and-daughter time when just for a couple of hours it was just us and I think we've really benefited from that' (ID A-072).

Participants spoke highly of the facilitators for the education sessions: 'Fantastic, very well presented, well engaging, knowledgeable' (ID A-058) and the physical activity sessions: 'they were excellent, really good lads… we were always busy, always sweating, it was all good' (ID B-089). Similarly, the facilitators spoke positively of the programme.

A key challenge was delivery over the UK winter:

The weather had been against us a lot of the times, I think if we had done it now in the summer, I think it would have been a lot better for us (ID B-089)

…although we've been out Saturdays and Sundays it's trying to fit it into darker nights when it's been cold and it's been wet or it's been snowing or whatever else, that's been difficult. (ID A-072)

## Progression criteria

Progression criteria for a full trial (table 4) were met for intervention acceptability to participants who attended and fidelity of delivery; recruitment to the trial was low and attrition was high (37.2%, n=16). Mean weight loss in the intervention arm was marginally below the prespecified ≥3 kg in the participants who were followed-up.

## DISCUSSION

The aims of the current study were to assess the feasibility of delivering a culturally adapted HDHK programme in a

| **Table 4** Progression criteria for a full trial | |
|---|---|
| **Progression criterion** | **Assessment** |
| Intervention is acceptable to a majority of fathers and families from differing black asian and minority ethnic groups and socioeconomic backgrounds. | Achieved |
| Randomisation occurs and more than 80% of those assessed accept randomisation. | Achieved |
| Recruitment of at least 68 out of the planned 90 fathers (75%) within the 4-month time frame. | 43 (48%) |
| Intervention implemented with fidelity in 75% of observations (see Process evaluation section). | Achieved |
| Attendance: 70% attending at least five of nine of the planned sessions. | 15 (52%) |
| More than 70% follow-up at 3 and 6 months. | 27 (62.8%) |
| Mean weight loss in the intervention arm of ≥3 kg. | 2.9 kg (95% CI −5.1 to 0.6) |

socioeconomically deprived UK setting and the feasibility of conducting a definitive RCT. Overall, it was possible to recruit and train facilitators to deliver the intervention programme with high fidelity. The programme was rated highly by the attending participants and delivery teams, and those who participated achieved weight loss comparable with other studies.[13 16] However, recruitment of participants was difficult, and aligning participant, venue and facilitator availability to deliver the programme was a major challenge. As such, participant recruitment and attendance rates were low and the progression criteria for the study were not met.

Despite multiple strategies, encouraging fathers to engage with the study was challenging. Attempts to enhance recruitment through schools, an approach used in the Australian setting, proved more difficult than anticipated. Schools are often involved in multiple projects and programmes, which may have meant they felt they did not have added capacity for HDHK-UK. It was recognised during recruitment that there can be sensitivities around discussing weight. To try to offset this, the opportunity for fathers to spend time with their children was emphasised. Challenges in recruitment have also been recognised in a similar study with fathers[37] and as a common theme in a systematic review of weight loss trials in men.[13 14] However, the findings and experiences from this current study differ from those of the Australian HDHK RCT.[16 17] Notably, the Australian study used a wait-list study design where all families were guaranteed to receive the programme at some point, which may have been more appealing for participants. An additional difference was a greater availability of sessions.

Nearly a third of participants allocated to the intervention group did not engage with the programme. This has been seen in other group-based health-related programmes[38–40] but differs from the HDHK trial in Australia.[16 17] The primary reason given for non-attendance or non-completion of the programme was that the session timings were not suitable, which was corroborated by the research team, which found identifying a convenient time for programme delivery to be one of the most significant challenges of the study. Evening sessions had to align to fathers' working patterns and children's school and bedtime routines, resulting in sessions often being set when traffic was at its heaviest. This was compounded by running the sessions in the winter with dark nights and poor weather. Weekend sessions often clashed with family activities, and venue and facilitator availability were more limited. Poor weather is likely to have also impacted recruitment, especially at schools where parents were not keen on stopping to talk to the study team in rainy or cold conditions. Delivery in summer and offering more choice of sessions might mitigate some of these challenges.

The facilitators reported it to be challenging to deliver the full content of the father's education session within the allocated 30 min, exacerbated by participants arriving late. The approach to sessions was to encourage interaction and discussion, which was appreciated by participants and necessary if the group size was small. However, this at times challenged the fidelity of delivery, and if the programme were to be rolled out, we would recommend reducing session content further.

The youngest children participating in the current study had difficulty in engaging with written materials; 4-year-olds were not eligible in the Australian studies.[16 17 40]

### Strengths and limitations

The study has a number of strengths. While successful weight management trials in men have been reported,[10 12 13 16 17] most participants in weight loss trials are women.[41 42] In addition, very few specifically target ethnic minority population groups[41] or recruit from socioeconomically disadvantaged populations, which is a unique contribution of this study.

This study also had limitations. First, the small number of participants recruited and attending the programme meant some adaptation for a smaller group size was required, but meant some of the group atmosphere was reduced. Those who attended enjoyed the programme, and while the remainder advised that session timing was their main reason for non-attendance, the small sample size created bias, especially for the qualitative interviews. While every attempt was made to interview non-attendees and non-completers, engagement was low. We had also planned to interview participants with their partners and children present, but logistically, this proved difficult to

arrange; many of the children were too young to participate in an interview, and most fathers opted for a phone interview as they found this more convenient.

Another limitation was the change in facilitation staff throughout the study, resulting in the need to run individualised training sessions after the original training delivered by the Fatherhood Institute. While the additional training was delivered by an experienced researcher (TLG), the process presented a significant time commitment and may have led to differences in delivery style.

Due to the study timeline and working with school terms and avoiding Ramadan, the intervention was delivered at the height of winter, a contrast to the Australian programme, which is delivered only in summer months. This is likely to have impacted both recruitment and intervention attendance.

The session observations and feedback from participants and facilitators were overwhelmingly positive. Despite the acknowledged limitations, the adapted HDHK programme was delivered and received well. The observed outcome measure of weight change across the programme was promising, and the data collected showed good levels of completion.

## Future research

At this time, the trial steering committee did not recommend progressing to a full RCT in the context in which this study was set. However, there may be other settings within the UK in which some of the barriers that were faced either would not occur or could not be addressed or ameliorated. Another avenue which could be explored would be for some of the education sessions to be delivered online, which has been shown to be successful in some groups[43] and would reduce the time difficulties experienced in delivering content in the face-to-face sessions. In line with the recommendations from the study patient and public involvement (PPI) group, delivery of the programme outside the context of weight management may result in higher recruitment.

## CONCLUSIONS

The majority of difficulties experienced in this study stem from the setting and context under which it was to be delivered, and while we do not recommend progressing this feasibility study to a full RCT, we recognise that the outcomes and recommendations made may have been different had the study been trialled in a different setting. A number of challenging circumstances converged within the study, which likely compounded further the difficulties encountered in delivery: recruitment difficulties, the untimely change in facilitation staff, and the poor weather throughout recruitment and delivery. The intervention was rated highly by those who attended, and the weight loss achieved by the intervention participants was promising, although no definitive conclusions can be drawn owing to the small sample size.

**Author affiliations**
¹Department of Health, University of Bath, Bath, UK
²Birmingham Clinical Trials Unit (BCTU), University of Birmingham, Birmingham, UK
³Health Services Management Centre, University of Birmingham, Birmingham, UK
⁴Institute of Applied Health Research, University of Birmingham, Birmingham, UK
⁵Fatherhood Institute, Marlborough, UK
⁶School of Health Sciences, University of Newcastle, Callaghan, New South Wales, Australia
⁷School of Sport, Exercise and Health Sciences, Loughborough University, Loughborough, UK
⁸Public member, Leamington Spa, UK
⁹Health Economics Unit, Institute of Applied Health Research, University of Birmingham, Birmingham, UK
¹⁰Cancer Research UK Clinical Trials Unit, University of Birmingham, Birmingham, UK
¹¹Birmingham Community Healthcare NHS Trust, Aston, UK
¹²School of Education, University of Newcastle, Callaghan, New South Wales, Australia

**Acknowledgements** We would like to thank all participating families and the local authorities for their support and direction and the facilitators who delivered the programme. We acknowledge members of the study steering committee: Professor John Wright, Bradford Institute for Health Research (Chair); Professor Pat Hoddinott, University of Stirling; Elaine Nicholls, Keele University; Mr Ray Fiveash, PPI representative. We acknowledge Kathy Jones and Cassius Campbell, who delivered the training programme, and Jeremy Davies from the Fatherhood Institute. We thank Jeszemma Garratt from the Fatherhood Institute and the Parent Advisory Panel and staff who contributed to the research delivery: Dr Khaled Ahmed, Felicity Brant, David Sardar and Meanaz Akhtar. We acknowledge academics from the University Newcastle, Australia, who developed the original Healthy Dads, Healthy Kids programme and who supported the study through the development of training materials and shared their experience of delivery in the Australian setting: Professor David Lubans and Kristen Saunders.

**Contributors** KJ, PA, AB, CC, AD, EF, LJ, MP, MS, MY and PM conceived the study; KJ was the principal investigator; YS undertook the statistical analysis; PH was the senior statistician; TG was the study coordinator and led the process evaluation; LJ was the qualitative lead; TG and KH undertook the qualitative interviews and analysis; PM and CC conceived and designed the original Healthy Dads, Healthy Kids intervention; PM and MY advised on training and delivery; CC and EM advised on dietary assessment; AE was the PPI lead; TG and KJ drafted the manuscript; all authors interpreted the findings, commented on paper drafts and agreed on the final version.

**Funding** Study funding was granted in October 2015 by the National Institute of Health Research (NIHR) Public Health Research programme (Ref 14/185/13); KJ is partly funded by NIHR Collaborations for Leadership and Health Research and Care West Midlands. The views expressed are those of the authors and not necessarily those of the NHS, the NIHR or the Department of Health and Social Care.

**Competing interests** PM and CC designed the original Healthy Dads, Healthy Kids Programme in Australia.

**Patient consent for publication** Not required.

**Ethics approval** Ethical approval for the two phases of the study was obtained from the University of Birmingham Science, Technology, Engineering and Mathematics Ethical Review Committee (16 January 2017, ethics reference; ERN_16–1323).

**Provenance and peer review** Not commissioned; externally peer reviewed.

**Data availability statement** Data are available upon reasonable request.

**ORCID iD**
Laura Jones http://orcid.org/0000-0002-4018-3855

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
