## [Reviewer comments · BMJ Open]

ARTICLE DETAILS

TITLE (PROVISIONAL)	A weight management programme for fathers: Healthy Dads, Healthy Kids UK – a feasibility RCT
AUTHORS	Griffin, Tania; Sun, Yongzhong; Sidhu, Manbinder S.; Adab, Peymane; Burgess, Adrienne; Collins, Clare; Daley, Amanda; Entwistle, Andrew; Frew, Emma; Hardy, Pollyanna; Hurley, Kiya; Jones, Laura; McGee, Eleanor; Pallan, Miranda; Young, Myles; Morgan, Philip; Jolly, Kate

VERSION 1 – REVIEW

REVIEWER	Enrique Rodilla Universidad Cardenal Herrera-CEU, CEU Universities, Hospital de Sagunto, Servicio de Medicina Interna, Spain
REVIEW RETURNED	30-Aug-2019

GENERAL COMMENTS	We have read with great interest this study, conducted to evaluate if a specific, validated weight-management program, adapted from the Healthy Dads Healthy Kids (HDHK), was feasible regarding a future RCT in a socioeconomically deprived and ethnically diverse UK setting. The introduction is clear, focusing on the burden of obesity in a large segment of the general population. The objective of the study is adequately described. The study design is appropriate to determine the aim of the study. It is very helpful to find a transparent assessment of predefined achievement criteria. In contrast to many other studies, the authors give an accurate description of HDHK, so the reader can understand perfectly the high quality of the intervention, and perhaps anticipate the difficulties in translating it into the practice. Primary and secondary outcomes are well characterized. The statistical methodology is adequate and justified. The presentation of the results is clear, the inclusion of original quotations of several participants adds to the quality of the study by helping to comprehend many if not all the difficulties the researchers encountered. Although negative, the results definitely answer the main questions addressed by the authors. The discussion contains the logical conclusions derived from the results, the authors stick to the pre-specified stop-go criteria and reasonably draw the conclusion that feasibility to progress to a RCT is not given. Strengths and limitations are included in this section. After reading such an accurate, ambitious, exigent and challenging paper, the only section that could possibly be improved is the paragraph dedicated to Future Research, as the authors transmit an impression of definite resignation, suggesting either to change the target (weight management) or the setting (socioeconomically deprived and ethnically diverse UK population). Some alternatives have recently been published, suggesting that internet-based platforms may be very helpful in changing unhealthy habits (Analysis
--

	of the efficacy of an internet-based self-administered intervention ("Living Better") to promote healthy habits in a population with obesity and hypertension: An exploratory randomized controlled trial. Mensorio MS et al. Int J Med Inform. 2019;124:13-23. doi: 10.1016/j.ijmedinf.2018.12.007.) It could be argued that internet may be not accessible to socially deprived populations, but this way to interact with the target population may increase the chance to reach obese fathers and their children, getting rid of facilitators and weather dependency. At least, it would be helpful if the authors could discuss this approach.
--	--

REVIEWER	Robert J Petrella Western University, Canada
REVIEW RETURNED	01-Sep-2019

GENERAL COMMENTS	This is a well written feasibility study of the adaptation of the Healthy Dads, Healthy Kids program in the UK by a very strong and large team among socioeconomically disadvantaged and ethnically diverse overweight/obese men and their 4-11 year old children in the West Midlands. Abstract: In the abstract the objectives should be two-fold (feasibility of delivering the HDHK program and feasibility of conducting an RCT) similar to the aims on page 19 first sentence of the Discussion. Outcomes appear to be on the men recruited however, Table 3 refers to "eldest child" for which there is no accompanying text or reflection in the Discussion. It would have been prudent to include such details and compare what was observed in Australia or in similar diads in the literature. In the "Strengths and limitations" page 3, I am not sure the first point is fully developed as a strength. The Introduction is complete but were there any competing interventions considered--what was the process involved in landing on the Heathy Dads Healthy Kids program--why not just focus on men alone as per Football Fans in Training? Were there any risks considered to participation rates when required to accommodate men and their chlidren--particularly in a setting of socioeconomic stressors? Methods: Why were the specific sites identified? Why was a formal sample size not considered given previous work in the field including Australian Healthy Dads, Healthy Kids to ensure recruitment was feasible? While the recruitment efforts appear very appropriate could you detail who performed the recruitment, how they were trained etc. It is notable that public involvement is described however how were they identified, were they representative of the general population etc? Did they have input into modifying or adding to the curriculum--did they advise against any specific outcome measures or recruiting methods? This would be quite helpful if considering re-tooling for other settings and interventions. You state that the "group" was facilitated by local, experienced and trained staff--please describe how this training was done and evaluated, what were the backgrounds of those involved? On page 6 you state that adaptations of Australian resources were made by one of the team (KJ) and one study partner--what about the public? The inclusion of a priori "progression criteria" is notable. What is the source of these criteria? Please include. Were materials available in other languages? On page 16 "challenges" it is not clear whether interim strategies were used and what the protocol was when rolling out recruitment--ie all at once "gorilla" approach vs step wise so as to not overburden
---

	or under burden potential targets. It is notable that facilitators reported time was a factor in not delivering all the desired content. Were there training sessions provided to deal with these scenarios or were facilitators trained to return to content in other ways or at subsequent sessions? Was some content less challenging than other? Discussion: It should be noted that both feasibility of delivery and feasibility of conducting an RCT were not met. While I would agree that wait list control design is appealing--how would this have changed recruitment in your study? Please expand on the issue of change in facilitation staff on page 21--I don't understand what specifically this impacted---I would think it may have been in your favour? Also, are you suggesting that should the number of circumstances been minimized that this would be a feasible delivery and feasible polo for an RCT?
--	--

REVIEWER	Linlin LI Department of Epidemiology and Biostatistics College of Public Health, Zhengzhou University Zhengzhou, People's Republic of China
REVIEW RETURNED	11-Sep-2019

GENERAL COMMENTS	Q1. The objective of this research is very novel, i'm interested with it, but the recruitment failed, so the results could not be acceptable. Q2. The results have no statistical analysis, there were no compare between 3 month and 6 month or HDHK-UK programme group and minimum intervention group, and no P-value and 95% confidence interval, so I don't know if the weight changes had any statistical significance and clinical value. Q3. the study was not powered to detect treatment effects on clinical outcomes, I think the study failed, the authors should recruit enough sample to do the study again.
--

VERSION 1 – AUTHOR RESPONSE

Reviewer 1: Enrique Rodilla

We thank the reviewer for their positive comments about the manuscript.

- Future Research:** The authors transmit an impression of definite resignation, suggesting either to change the target (weight management) or the setting (socioeconomically deprived and ethnically diverse UK population). Some alternatives have recently been published, suggesting that internet-based platforms may be very helpful in changing unhealthy habits (Analysis of the efficacy of an internet-based self-administered intervention ("Living Better") to promote healthy habits in a population with obesity and hypertension: An exploratory randomized controlled trial. Mensorio MS et al. Int J Med Inform. 2019;124:13-23. doi: 10.1016/j.ijmedinf.2018.12.007.) It could be argued that internet may be not accessible to socially deprived populations, but this way to interact with the target population may increase the chance to reach obese fathers and their children, getting rid of facilitators and weather dependency. At least, it would be helpful if the authors could discuss this approach.

We have added the consideration of utilising the internet for delivery of some content of HDHK in the 'Future research' section:

Another avenue which could be explored would be for some of the education sessions to be delivered online which has been shown to be successful in some groups⁴⁵ and would reduce the time difficulties experienced in delivering content in the face to face sessions.

Reviewer 2: Robert J Petrella

- 1. Abstract: In the abstract the objectives should be two-fold (feasibility of delivering the HDHK program and feasibility of conducting an RCT) similar to the aims on page 19 first sentence of the Discussion.**

This has now been included in the abstract, which now reads:

Objectives: To assess (i) the feasibility of delivering a culturally adapted weight management programme; Healthy Dads, Healthy Kids United Kingdom (HDHK-UK) for fathers with overweight or obesity and their primary school aged children, and (ii) the feasibility of conducting a definitive randomised controlled trial.

- 2. Outcomes appear to be on the men recruited however, Table 3 refers to "eldest child" for which there is no accompanying text or reflection in the Discussion. It would have been prudent to include such details and compare what was observed in Australia or in similar diads in the literature.**

We have kept the description of the children to a minimum as the main focus of the study was the men. We have added some further reference to the children in the results. We have not added discussion about the findings of the children, nor drawn comparison with the Australian studies' findings as that would require considerably more text, and we feel it is important to keep the paper concise and focused on the main objectives of the intervention, which relate to weight management in men.

However to add clarity to the manuscript we have added two short additions to the beginning of the results section as follows (additions underlined) (pages 8-9):

Despite multiple recruitment methods, only 92 men expressed interest in the study, of whom 43 were recruited and randomised (intervention group n=29) (Figure 1); 62 children were recruited.

The mean age of their participating children (n=62) was 7.7 years (range 4.0–11.7) of whom 20 (32.8%) were overweight or obese.

- 3. In the "Strengths and limitations" page 3, I am not sure the first point is fully developed as a strength.**

To ensure the strength of the first bullet point has been reworded and clarity added to emphasise how we consider this point to be a strength (page 3):

This unique study provides new evidence into the feasibility of delivery of a weight management programme targeting fathers from socio-economically deprived, ethnically diverse community settings, and its acceptability in this target population, a key strength given the relative paucity of research in this area.

- 4. The Introduction is complete but were there any competing interventions considered--what was the process involved in landing on the Healthy Dads Healthy Kids program--why not just focus on men alone as per Football Fans in Training? Were there any risks considered to participation rates when required to accommodate men and their children--particularly in a setting of socioeconomic stressors?**

Thank you for this interesting question. The key part of the Healthy Dads Healthy Kids programme is the involvement of children and family as part of the approach to lifestyle change.

Using football clubs as a setting has proven successful in a subset of men who are typically middle-age and less ethnically diverse, but we were interested in younger men within the family context so this study was specifically designed to be offered in a community setting and involve children as a way to motivate the fathers. The leisure and community centre setting is accessible to all and is not restricted to those interested in a specific sport.

Our UK public health programmes are generally targeted at areas of socio-economic disadvantage. There are risks to participation rates in these settings, but we believe that it is crucial to undertake research in the populations who would have most to gain from public health interventions, which in turn feeds into the overall key public health aim of reducing health inequalities. We undertook considerable formative work prior to this study to determine the acceptability of the programme.

- 5. Methods: Why were the specific sites identified? Why was a formal sample size not considered given previous work in the field including Australian Healthy Dads, Healthy Kids to ensure recruitment was feasible?**

Whilst the original HDHK has shown to be feasible and successful in Australia, this study involved the cultural adaption of the intervention and delivering the trial in a socioeconomically deprived UK setting. Before a large scale RCT could be conducted, with a formally calculated sample size, it was important to test the feasibility of the intervention and the processes which would be involved in an RCT (study recruitment, consent and follow-up, ability to deliver intervention, intervention fidelity, adherence and acceptability, weight loss, using questionnaires and measurements at baseline, 3 and 6-months, and through qualitative interviews).

We have added some additional text as to why the two sites were selected (page 4):

It was conducted in two urban local authority areas of the West Midlands, UK, (site 1 and site 2) selected for their population profile and interest in supporting the programme.

- 6. While the recruitment efforts appear very appropriate could you detail who performed the recruitment, how they were trained etc.**

The research team recruited participants. They all hold extensive experience of recruitment of participants to trials in community settings. This has been clarified in the text as follows (page 5):

Fathers were recruited (September 2017-January 2018) by the research team who had extensive experience of participant recruitment in a community setting.

- 7. It is notable that public involvement is described however how were they identified, were they representative of the general population etc? Did they have input into modifying or adding to the curriculum--did they advise against any specific outcome measures or recruiting methods? This would be quite helpful if considering re-tooling for other settings and interventions.**

The PPI group was brought together by the Fatherhood Institute who were partners in this study. It would be hard to confirm how representative a group of 10 people are to a local population, so we haven't added this detail. The group was ethnically diverse and lived and worked in one of the localities. We have added some more detail to the section of PPI (page 8):

A public and patient participation group of eight fathers and two mothers from one of the research sites were involved throughout all stages of the study by contributing to decisions about outcome measures, commenting on intervention materials (in particular the need for their simplification), and participant facing documents (downplaying the focus on weight management and focussing on the opportunity for father-child interaction in recruitment materials), advising on recruitment strategies and dissemination of the findings.

- 8. You state that the "group" was facilitated by local, experienced and trained staff--please describe how this training was done and evaluated, what were the backgrounds of those involved?**

To ensure this is clear in the manuscript, the following two sentences have been included (pages 5-6):

Facilitators were selected based on their experience of delivering group programmes and delivering health advice; they included health trainers (who provide community support for health related behaviour changes) and sports coaches. They completed HDHK delivery training with either the Fatherhood Institute or the research team. The training included practicing delivery of parts of the intervention.

- 9. On page 6 you state that adaptations of Australian resources were made by one of the team (KJ) and one study partner--what about the public?**

The PPI group identified the need for adaptations to the HDHK programme and we then undertook an adaptation phase with qualitative interviews informing the need for adaptations. This is described comprehensively in the NIHR Public Health Research journal, which is in press. We have added some more detail to the section about the adaptation and referenced the NIHR report (page 6):

Adaptations to the Australian resources were made by the research team (KJ and MS) and one of the study partners (The Fatherhood Institute) in conjunction with the wider research team. The adaptations were informed by qualitative research with fathers and mothers from similar socio-economic backgrounds and residing in the same geographical region. Adaptations focussed on reducing the number of PowerPoint slides, simplifying and anglicising wording and updating the guidance and statistics to align to UK public health recommendations. References to foods, activities and images were updated to reflect a multi-cultural UK population. Fathers and their children attended every session and mothers were invited to one session when family food was discussed.

- 10. The inclusion of a priori "progression criteria" is notable. What is the source of these criteria?**

These were developed in the research proposal and agreed by the funding panel and study steering group. The section now reads (page 7):

Progression criteria, agreed by the funding panel and Study Steering Committee, were pre-defined to help evaluate whether the feasibility trial should be recommended to progress to a fully powered RCT. These are detailed in the results section.

11. Were materials available in other languages?

The materials were only available in English. In the section 'Study design and participants' it states men were not eligible to take part if they "were unable to speak and/or understand English".

12. On page 16 "challenges" it is not clear whether interim strategies were used and what the protocol was when rolling out recruitment--ie all at once "gorilla" approach vs step wise so as to not overburden or under burden potential targets.

With a limited number of people recruiting to the study a step-wise approach was taken. We have tried to clarify this by adding to the methods section that the strategies were used over a period (page 5):

A range of methods were used over the recruitment period...

13. It is notable that facilitators reported time was a factor in not delivering all the desired content. Were there training sessions provided to deal with these scenarios or were facilitators trained to return to content in other ways or at subsequent sessions? Was some content less challenging than other?

To clarify this we have added further detail as follows (page 18):

The intervention was delivered with high fidelity. The style of delivery and the small group sizes meant group discussions were encouraged, allowing content to be tailored to the group's needs (facilitators were trained to manage group discussions as part of the HDHK delivery training); however, facilitators reported finding it challenging to deliver all the session content in the allocated time. This was especially difficult if participants arrived to the sessions late and facilitators had to balance content delivery whilst also allowing for group discussion and interaction.

14. Discussion: It should be noted that both feasibility of delivery and feasibility of conducting an RCT were not met. While I would agree that wait list control design is appealing--how would this have changed recruitment in your study?

The guarantee of a programme may have helped to study be a more appealing as the burden taking part in all three sets of measurements with no guarantee of the programme was likely unappealing to some. We have stated this in the discussion:

..the Australian study used a wait-list study design where all families were guaranteed to receive the programme at some point, which may have been more appealing for participants.

15. Please expand on the issue of change in facilitation staff on page 21--I don't understand what specifically this impacted---I would think it may have been in your favour?

The change in facilitation staff gave rise to the need for additional training and thus additional time and cost. It also meant that the training was delivered by different trainers resulting in less consistency. We have explained this in the paragraph in the limitation section:

Another limitation was the change in facilitation staff throughout the study, resulting in the need to run individualised training sessions after the original training delivered by the Fatherhood Institute. Whilst the additional training was delivered by an experienced researcher (TG), the process presented a significant time commitment and may have led to differences in delivery style.

16. Are you suggesting that should the number of circumstances been minimized that this would be a feasible delivery and feasible polo for an RCT?

Yes, there were many circumstantial challenges experienced in this study which have been outlined in this paper. Whilst this is the nature of research, we feel there were a number of turning points throughout the study, which if by circumstance had not been present the outcome could have differed considerably.

Reviewer 3: Linlin LI

1. The objective of this research is very novel, I'm interested with it, but the recruitment failed, so the results could not be acceptable.

We agree with this reviewer that interpretation of the outcome measures of a future trial need to be interpreted with great caution, However, as this was a feasibility study, ability to recruit to the study was a key outcome of interest and is one of the progression criteria; therefore not achieving recruitment targets is in itself an important finding.

2. The results have no statistical analysis, there were no compare between 3 month and 6 month or HDHK-UK programme group and minimum intervention group, and no P-value and 95% confidence interval, so I do not know if the weight changes had any statistical significance and clinical value.

This was not a study designed to assess effectiveness, and hence, in line with best practice for feasibility trials, we have not undertaken between group statistical testing or hypothesis testing. We pre-specified this is the statistical analysis plan that was agreed by the study steering committee.

3. The study was not powered to detect treatment effects on clinical outcomes, I think the study failed, the authors should recruit enough sample to do the study again.

The study was not designed to be powered for the detection of any treatment effects and therefore the research is not powered to make a statistical comparison between the control and intervention group. As a feasibility study, it was powered to estimate recruitment, follow-up and

questionnaire completion rates to within +/- 10% with 95% confidence, based on a worst case estimate of 50%.

VERSION 2 – REVIEW

REVIEWER	Enrique Rodilla Hospital de Sagunto Universidad Cardenal Herrera-CEU, CEU Universities Valencia, Spain
REVIEW RETURNED	24-Oct-2019
GENERAL COMMENTS	The authors have answered adequately to all the queries.